

# Integrated analysis of the transcriptome-wide m6A methylome in preeclampsia and healthy control placentas

Jin Wang[1], Fengchun Gao[2], Xiaohan Zhao[1], Yan Cai[1] and Hua Jin[1]

[1] Prenatal Diagnosis Center, Jinan Maternal and Child Health Care Hospital, Jinan, China
[2] Obstetrical Department, Jinan Maternal and Child Health Care Hospital, Jinan, China

## ABSTRACT

N6-methyladenosine (m6A) is the most prevalent modification in eukaryotic mRNA and potential regulatory functions of m6A have been shown by mapping the RNA m6A modification landscape. m6A modification in active gene regulation manifests itself as altered methylation profiles. The number of reports regarding to the profiling of m6A modification and its potential role in the placenta of preeclampsia (PE) is small. In this work, placental samples were collected from PE and control patients. Expression of m6A-related genes was investigated using quantitative real-time PCR. MeRIP-seq and RNA-seq were performed to detect m6A methylation and mRNA expression profiles. Gene ontology (GO) functional and Kyoto encyclopedia of genes and genomes (KEGG) pathway analyses were also conducted to explore the modified genes and their clinical significance. Our findings show that METTL3 and METTL14 were up-regulated in PE. In total, 685 m6A peaks were differentially expressed as determined by MeRIP-seq. Altered peaks of m6A-modified transcripts were primarily associated with nitrogen compound metabolic process, positive regulation of vascular-associated smooth muscle cell migration, and endoplasmic reticulum organisation. The m6A hyper-methylated genes of Wnt/$\beta$-catenin signalling pathway, mTOR signalling pathway, and several cancer-related pathways may contribute to PE. We also verified that the significant increase of *HSPA1A* mRNA and protein expression was regulated by m6A modification, suggesting m6A plays a key role in the regulation of gene expression. Our data provide novel information regarding m6A modification alterations in PE and help our understanding of the pathogenesis of PE.

# INTRODUCTION

Preeclampsia (PE) is a multi-system disorder that is primarily characterised by new-onset hypertension accompanied by proteinuria during gestation. This disease affects 3–5% of all pregnancies and is one of the leading causes of maternal and perinatal morbidity and mortality (*Mol et al., 2016*). The exact pathophysiology that causes PE remains unclear; however, genetic, immunological, endocrine, and environmental factors have been implicated in its pathogenesis (*Burton et al., 2019*). The placenta plays an essential role in the development of PE. The underlying pathogenic mechanisms include defective

Corresponding author
Hua Jin, wangjinshiyi@126.com

deep placentation, oxidative and endoplasmic reticulum stress, intravascular inflammation, and imbalance of angiogenesis, among others (*Burton et al., 2019*; *Chaiworapongsa et al., 2014*). No specific treatment is currently available, and delivery of the placenta is the only effective treatment (*Burton et al., 2019*; *Chaiworapongsa et al., 2014*; *Wang et al., 2019*).

Numerous studies have reported that PE significantly alters the expression of coding and noncoding RNAs (ncRNAs), including mRNA, miRNA, long noncoding RNA (lncRNA), and circular RNA (circRNA) (*Bai et al., 2018*; *Liu et al., 2017b*; *Muller-Deile et al., 2018*; *Nikuei et al., 2017*). However, while these studies explored RNA expression, the modification profiles of these RNAs in the context of PE remain to be characterised. N6-methyladenosine (m6A) modification in mRNA is prevalent, and functionally modulates in eukaryotes that is mediated by the m6A methyltransferase complex, methyltransferase-like (METTL)3, METTL14, and Wilms' tumour 1-associated protein (WTAP) and eliminated by fat-mass and obesity-associated protein (FTO) or alkylation repair homolog protein 5 (ALKBH5) (*Boccaletto et al., 2018*; *Fu et al., 2014*; *Meyer & Jaffrey, 2014*). These modifications are believed to moderate RNA structure, function, and stability (*Edupuganti et al., 2017*; *Liu et al., 2017a*; *Piao, Sun & Zhang, 2017*; *Wang et al., 2014*). Recently, the effects of m6A modification on many fundamental biological processes have been characterised; these processes include metabolism (*Yang et al., 2018*), immunomodulation (*Zheng et al., 2017*), carcinogenesis(*Ma et al., 2017*; *Zhang et al., 2017a*), and spermatogenesis (*Chen et al., 2017*), among others. Abnormal m6A methylation is associated with a variety of human diseases, such as obesity, neuronal disorders, cancer, and infertility (*Chen et al., 2017*; *Ma et al., 2017*; *Yang et al., 2018*; *Zhang et al., 2017a*; *Zheng et al., 2017*).

Given the indispensable function of RNA m6A modification in various bioprocesses, it is reasonable to speculate that deregulation of m6A modification may also be associated with PE. A recent study indicate that m6A at 5′-UTR and nearby stop codon in placental mRNA may play important roles in fetal growth and PE through conducted MeRIP-Seq on human placentas obtained from mothers of infants of various birth weights (*Taniguchi et al., 2020*). Here, we aimed to compare the m6A-tagged transcript profiles of placentas from PE-affected pregnancies with those of placentas from healthy pregnancies to identify gene-specific changes in RNA methylation that may regulate placental gene expression and contribute to the development of PE. Additionally, potential roles for the m6A-modified transcripts in the physiological and pathological mechanisms underlying PE were revealed; these can provide a theoretical basis for the prevention and pre-emptive treatment of PE.

## MATERIALS & METHODS

### Sample collection

All placental samples used in this study were collected from the Department of Obstetrics, Jinan Maternal and Child Health Care Hospital. The characteristics of puerperants and newborns were collected from hospital medical records. PE cases ($n = 4$) were collected in light of the guideline designed by the American College of Obstetrics and Gynecology (ACOG) (*ACOG Practice Bulletin, 2019*; *Magee et al., 2014*). These cases included women

**Table 1  Clinical information of samples used in the study.**

| Characteristic | Preeclampsia (*n* = 4) | Control (*n* = 4) | *P* value |
|---|---|---|---|
| Maternal age, year | 29.5 ± 1.3 | 29 ± 0.8 | 0.54 |
| Gestational age, week | 37.9 ± 0.5 | 38.2 ± 0.4 | 0.41 |
| Prenatal maternal body mass index, kg/m$^2$ | 27.6 ± 1.2 | 27.7 ± 0.8 | 0.86 |
| Diastolic pressure, mmHg | 110.3 ± 7.1 | 80 ± 4.2 | 3.40E−04 |
| Systolic pressure, mmHg | 162.5 ± 4.8 | 124.5 ± 4.9 | 3.27E−05 |
| Proteinuria, g/24 h | 4.1 ± 0.5 | 0 ± 0 | 4.08E−04 |
| Birth weight, g | 3567.3 ± 290.2 | 3339 ± 316.6 | 0.33 |

who exhibited a blood pressure of ≥140/90 mmHg on two occasions that occurred at least 4 h apart, accompanied by proteinuria (2+ on dipstick or 300 mg/24 h) at ≥20 weeks and <34 weeks of gestation. Normal controls (*n* = 4) were defined as pregnancies without PE. All participants were of Han Chinese descent and those with complications such as gestational hypertension, gestational diabetes mellitus, foetal growth restriction, or preterm birth (<37 weeks) were excluded (Table 1). The study protocols were approved by the Ethics Review Committee of Ji'nan Maternal and Child Health Care Hospital and conducted in accordance with the Declaration of Helsinki (No. JNFY-2019003).

Placental samples were collected at three sites from the foetal side of the placenta immediately after caesarean section (<30 min). Fragments of approximately 1 cm$^3$ were dissected from the placenta after removing maternal blood by vigorous washing in ice-cold saline; these fragments were snap-frozen in liquid nitrogen and stored at −80 °C until use. All participants provided written informed consent prior to participation.

## RNA isolation

The placental samples from three different sites were ground and mixed, then, total RNA was isolated using TRIzol reagent (Invitrogen, CA, USA) according to the manufacturer's protocol. Agarose gel electrophoresis and NanoDrop ND-1000 (Thermo Fisher Scientific, MA, USA) was used to monitor RNA integrity and quality. Intact mRNA was isolated from the total RNA samples using an Arraystar Seq-Star$^{TM}$ poly(A) Mrna Isolation Kit in accordance with the manufacturer's protocol.

## Quantitative real-time PCR

The expression of m6A-related genes *METTL3*, *METTL14*, *FTO*, *WTAP*, and *ALKBH5* were detected by quantitative real-time PCR (RT-qPCR). Briefly, total RNA was isolated using TRIzol reagent (Invitrogen), and cDNA was generated by reverse transcription using Prime Script $^{TM}$RT Master Mix (Perfect Real-Time; Takara Bio, Shiga, Japan). RT-PCR was performed using SYBR Green master mix (Yeasen, Shanghai, China) and a thermal cycler (LightCycler System; Roche Diagnostics Corp, IN, USA). *β*-Actin was used as an internal control to normalise the data. The primers used for RT-qPCR are presented in Table 2.
**Table 2** Sequences of primers used for qRT-PCR analysis of mRNA levels.

| Gene name | Primer | Sequence | Product size (bp) |
|---|---|---|---|
| *METTL3* | Forward | 5′ACAGAGTGTCGGAGGTGATT 3′ | 201 |
| | Reverse | 5′TGTAGTACGGGTATGTTGAGC 3′ | |
| *METTL14* | Forward | 5′TGAGATTGCAGCACCTCGAT 3′ | 250 |
| | Reverse | 5′AATGAAGTCCCCGTCTGTGC 3′ | |
| *WTAP* | Forward | 5′CCTCTTCCCAAGAAGGTTCGAT 3′ | 238 |
| | Reverse | 5′GTTCCTTGGTTGCTAGTCGC 3′ | |
| *FTO* | Forward | 5′AATAGCCGCTGCTTGTGAG 3′ | 182 |
| | Reverse | 5′CCACTTCATCTTGTCCGTTG 3′ | |
| *ALKBH5* | Forward | 5′GCCGTCATCAACGACTACCA 3′ | 208 |
| | Reverse | 5′ATCCACTGAGCACAGTCACG 3′ | |
| *HSPA1A* | Forward | 5′CCACCATTGAGGAGGTAGATTAG3′ | 176 |
| | Reverse | 5′CTGCATGTAGAAACCGGAAAA3′ | |
| *β-actin* | Forward | 5′GTGGCCGAGGACTTTGATTG 3′ | 73 |
| | Reverse | 5′CCTGTAACAACGCATCTCATATT 3′ | |

## Quantification of m6A in total RNA

The m6A RNA methylation status was directly detected using the EpiQuik[TM] m6A RNA Methylation Quantification Kit (Colorimetric) according to the manufacturer's protocol. Briefly, a negative control and a standard curve consisting of six different concentrations (range: from 0.02 to 1 ng of m6A) were prepared. Two hundred nanograms of total RNA was used for each reaction. After RNA binding to the 96-well plates, the binding solution was removed and the plates were washed three times with diluted wash buffer. Then, diluted capture anti-m6A antibodies were added; subsequently, the plates were washed four times with diluted wash buffer. Add 100 µl of developer solution to each well and incubate at room temperature for 7 min away from light. Add 100 ul of stop solution to each well to stop enzyme reaction. The optical density (OD) at 450 nm was measured using a microplate reader (BIOTEK, Vermont, USA). The absolute amount of m6A was quantified, and the percentage of m6A within the total RNA was calculated.

## m6A-RIP-seq and data analysis

Poly(A) RNA was extracted with Arraystar Seq-Star[TM] poly(A) mRNA Isolation Kit (Arraystar, MD, USA). The RNA was fragmented into fragments with an average length of 100 nt using RNA Fragmentation Reagents (Sigma, MO, USA). Fragmented mRNAs were incubated for 2 h at 4 °C in the presence of 2 µg m6A antibodies (Synaptic Systems, 202003) in a 500 µl IP reaction system, and some of the fragments were used as input. The mixture was then incubated with protein-A beads and purified using elution buffer and ethanol. RNA-seq libraries for m6A antibody-enriched mRNAs and input mRNAs were prepared using the KAPA Stranded mRNA-seq Kit (Illumina, CA, USA). Finally, the completed libraries were assessed using an Agilent 2100 Bioanalyzer. The libraries were denatured with 0.1M NaOH and loaded into the reagent cartridge. Clusters were generated

using an Illumina cBot system (#PE-410-1001, Illumina). Sequencing was performed on an Illumina HiSeq 4000 machine following HiSeq 3000/4000 SBS Kit (300 cycles) protocols.Quality control of the sequence data was performed using FastQC (v0.11.7). The raw data were trimmed using Trimmomatic software (v0.32) and aligned to the Ensembl reference genome using HISAT2 software (v2.1.0). The m6a-RIP-enriched regions (peaks) were detected using exomePeak software (v2.13.2). The differential m6A peaks (fold changes $\geq$1.5 and $p < 0.05$) between the case group and control group were analysed using exomePeak. These differential peaks were annotated using the Ensembl database (GRCh 37/hg19). DREME motif discovery in transcription factor ChIP-seq data was used to identify motifs among the m6A peak sequences. Gene ontology (GO) terms and Kyoto encyclopedia of genes and genomes (KEGG) pathways were analyzed through GO database and KEGG pathway database.

## RNA-Sequencing

Intact mRNA was isolated from total RNA samples using a NEBNext Poly(A) mRNA Magnetic Isolation Module (New England Biolabs, Hertfordshire, UK) according to the manufacturer's protocol. RNA-Seq libraries were prepared using a KAPA Stranded RNA-Seq Library Prep Kit (Illumina). Sequencing was performed using the Illumina HiSeq 4000 platform.

## Quantifies m6A levels of *HSPA1A* using MazF-qPCR

A previously reported strategy was adopted to identify m6A levels using the methyl-sensitivity of MazF RNA endonuclease, which was previously shown to cleaves RNA at unmethylated ACA sites, but not at sites with m6A methylation (*Zhang et al., 2019*). 500ng RNA was denatured by heating at 70 °C for 2 min, then placed on ice. Each tube was supplemented with 2 μl 5x MazF buffer, 1 μl mRNA interferase-MazF(TakaRa, 20U/ul), 1 μl RNase inhibitor, and DEPC-treated water was added to the total volume of 10 ul. After mixing, the mixtures were incubated at 37 °C for 30 min and put on ice for later use. MazF digested RNA (MazF +) and RNA without digestion (MazF -) was reverse-transcribed into cDNA using Prime ScriptTM RT Master Mix (Takara Bio, Shiga, Japan). RT-qPCR adopted the same method as described above, and the data were analyzed by $2^{-\Delta\Delta CT}$ method. The m6A methylation at that particular site of *HSPA1A* was the expression of *HSPA1A* with MazF treatment, which was normalized by MazF -. *HSPA1A* primer set: Fwd: 5′ AGGCCGACAAGAAGAAGGTG3′, Rev: 5′CCTGGTACAGTCCGCTGATG3′.

## RT-qPCR of *HSPA1A* mRNA

*HSPA1A* mRNA was quantified by RT-qPCR, which adopted the same method as described above, and the primers used for RT-qPCR are presented in Table 2.

## Western blot analysis

Western blot (WB) was performed in 4 PE samples and 4 control samples using placental tissues. Briefly, protein was extracted from placental samples by the methods of TRIzol. Block the membrane with 5% skim milk in Tris-Buffered Saline Tween-20 (TBST) for 1 h. Primary antibody was used in 5% bovine serum albumin (BSA) and incubated at room

temperature for 2 h, then washed the membrane with TBST for 4 times, 8 min each time. The secondary antibody was incubated for 1.5 h, and the membrane was washed with TBST for 3 times. Membrane blots were visualized using ECL plus detection kit (Thermo Fisher Scientific) and imaged on ImageQuant 4600 (Tenon, Shanghai, China).

## Statistical analyses

Data are expressed as the mean ± standard deviation (SD). All statistical analyses were conducted using the SPSS 22.0 statistical package. Student's $t$-test was used to compare m6A, mRNA and protein levels between the PE and control samples. Fisher's exact test was used for all bioinformatic analyses, and a $p$-value < 0.05 was considered statistically significant.

# RESULTS

## The main clinical information of samples

The main clinical data of the PE cases and controls are summarised in Table 1. All women did not have diabetes, chronic hypertension, polycystic ovarian syndrome, kidney or liver disease, and serious infection. Blood pressure and proteinuria were significantly higher in the PE cases than in the controls. There was no significant difference in maternal age, gestational age, prenatal maternal body mass index, and birth weights at delivery between the groups.

## METTL3 and METTL14 were up-regulated in PE

The concentration, purity and integrity of extracted RNA was shown in Table S1 and Fig. S1. Using qRT-PCR, we examined the mRNA levels of five core enzymes responsible for m6A modification, including METTL3, METTL14, FTO, WTAP, and ALKBH5, in the PE and control samples. mRNA levels of METTL3 and METTL14, the key m6A methyltransferase, were significantly increased in the PE samples compared to the control (Fig. 1; METTL3 $p = 0.023$; METTL14 $p = 0.016$). WTAP and the erasers FTO and ALKBH5 were not significantly dysregulated in the PE group (Fig. 1).

## m6A modification profiles in PE

In this study, each sample was generated nearly 20 M m6A-RIP-seq reads, and the quality control metrics of the sequence data are shown in Table S2. The level of m6A RNA methylation within the total RNA in the PE was higher than that in the controls (Fig. 2). We then analysed the genome-wide profiling of the m6A-modified mRNA in the PE and control samples (GEO accession number: GSE143966). In total, 370 m6A peaks were significantly up-regulated, whereas 315 peaks were down-regulated (fold changes ≥1.5 and $p < 0.05$; Fig. 3A). The top 20 altered methylated m6A peaks are listed in Table 3, and all significantly differentially expressed m6A peaks are listed in Table S3. Additionally, 203 genes had up-regulated m6A peaks, and 202 genes had down-regulated m6A peaks (fold changes ≥1.5 and $p < 0.05$; Table S3). Each gene may have one or more modified peaks. The identified m6A peaks were primarily enriched within the coding sequence in proximity to the stop codons and in the 3′UTR (Figs. 3B–3D). Additionally, the m6A peaks

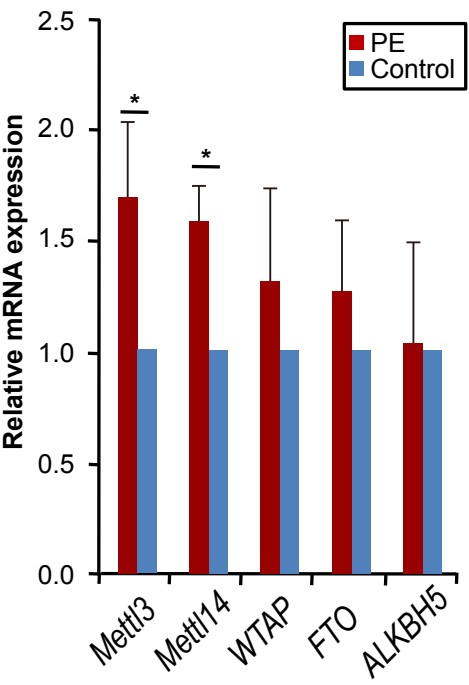

**Figure 1** *METTL3* and *METTL14* were up-regulated in PE. Quantitative real-time PCR was used to analysis the mRNA levels of *METTL3, METTL14, FTO, WTAP*, and *ALKBH5* in the PE samples and samples from normal pregnant women. All $p$-values were calculated using Student's $t$-test. *$p < 0.05$ versus control group ($n = 4$ each). PE, preeclampsia.

were characterised by the canonical RRACH motif (R represents purine, A is m6A, and H is a non-guanine base; Fig. 3E). *HSPA1A*, a representation of the significantly up-regulated peak, is shown in Figs. 3F, 3G and Fig. S2.

## GO analysis and pathway analysis of differentially methylated mRNA

To investigate the functional physiological and pathological significance of m6A modification in PE, GO functional analysis and KEGG pathway analysis were used to examine the altered m6A peaks. GO analysis (the count of genes involved in a GO term >2, $p < 0.05$) revealed that the up-regulated peaks in PE were significantly involved in macromolecule metabolic processes and the maintenance of DNA repeat elements, nuclear and intracellular processes, and damaged DNA binding and heat shock protein binding (Fig. 4A). The down-regulated peaks were significantly associated with organelle organisation and membrane docking, intracellular and intracellular organelle functions, and damaged DNA binding and cysteine-type endopeptidase activity involved in the execution phase of apoptosis (Fig. 4B).

KEGG pathway analysis demonstrated that the up-regulated peaks in the PE group were significantly associated with the Wnt signalling pathway, the mTOR signalling pathway, and the AMPK signalling pathway (Fig. 4C). The down-regulated peaks were significantly associated with amino sugar and nucleotide sugar metabolism (Fig. 4D).

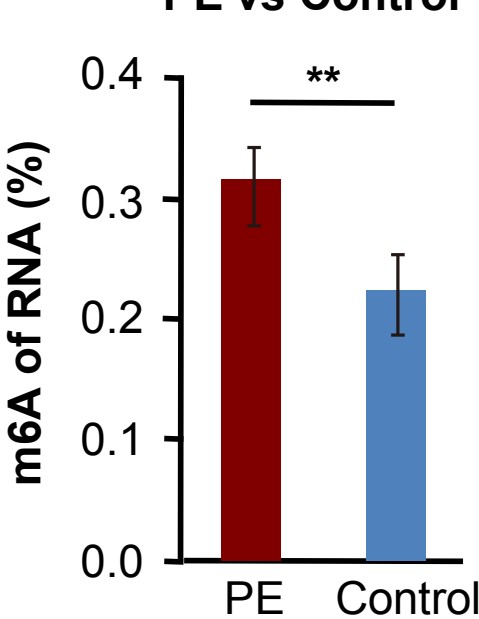

**Figure 2** **m6A levels of total RNA in PE and control.** m6A levels of total RNA were determined by antibody based colorimetric method. Data are presented as mean ± SD. All *p*-values were calculated using Student's *t*-test. **$p < 0.01$ versus control group ($n = 4$ each). PE, preeclampsia.

## Overview of transcriptome profiles and conjoint analysis of m6A-RIP-seq and RNA-seq data

In this study, each sample was generated nearly 20 M RNA-seq reads, and the quality control metrics of the RNA-seq data are shown in Table S2. Transcriptome profiles of the altered genes in PE were determined by RNA-seq (GEO accession number: GSE143953). Significantly differentially expressed genes (fold change ≥1.5 and $p < 0.05$) between the PE and control samples were identified using a volcano plot (Fig. 5A); these included 68 up-regulated and 52 down-regulated genes. The top twenty altered genes are listed in Table 4. The top ten GO and pathways associated with the up- or down-regulated genes are displayed in Fig. S3. Hierarchical cluster analysis was performed to identify differentially expressed genes between the two groups (Fig. 5B). According to conjoint analysis of m6A-RIP-seq and RNA-seq data, all genes were initially divided into four groups that included 55 hyper-methylated m6A peaks in mRNA transcripts that were significantly up-regulated (30; hyper-up) or down-regulated (25; hyper-down) and 50 hypo-methylated m6A peaks in mRNA transcripts that were significantly up-regulated (34; hypo-up) or down-regulated (16; hypo-down) (Fig. 5D, Table S4).

## The expression of *HSPA1A* is regulated by m6A modification

In this study, we detected the m6A levels of *HSPA1A* at the particular site in the CDS region were significantly upregulated in PE (Figs. 6A & 6B; $p = 0.012$). The particular site include oligos "ACA" and "(m6A)CA", and the sequences were
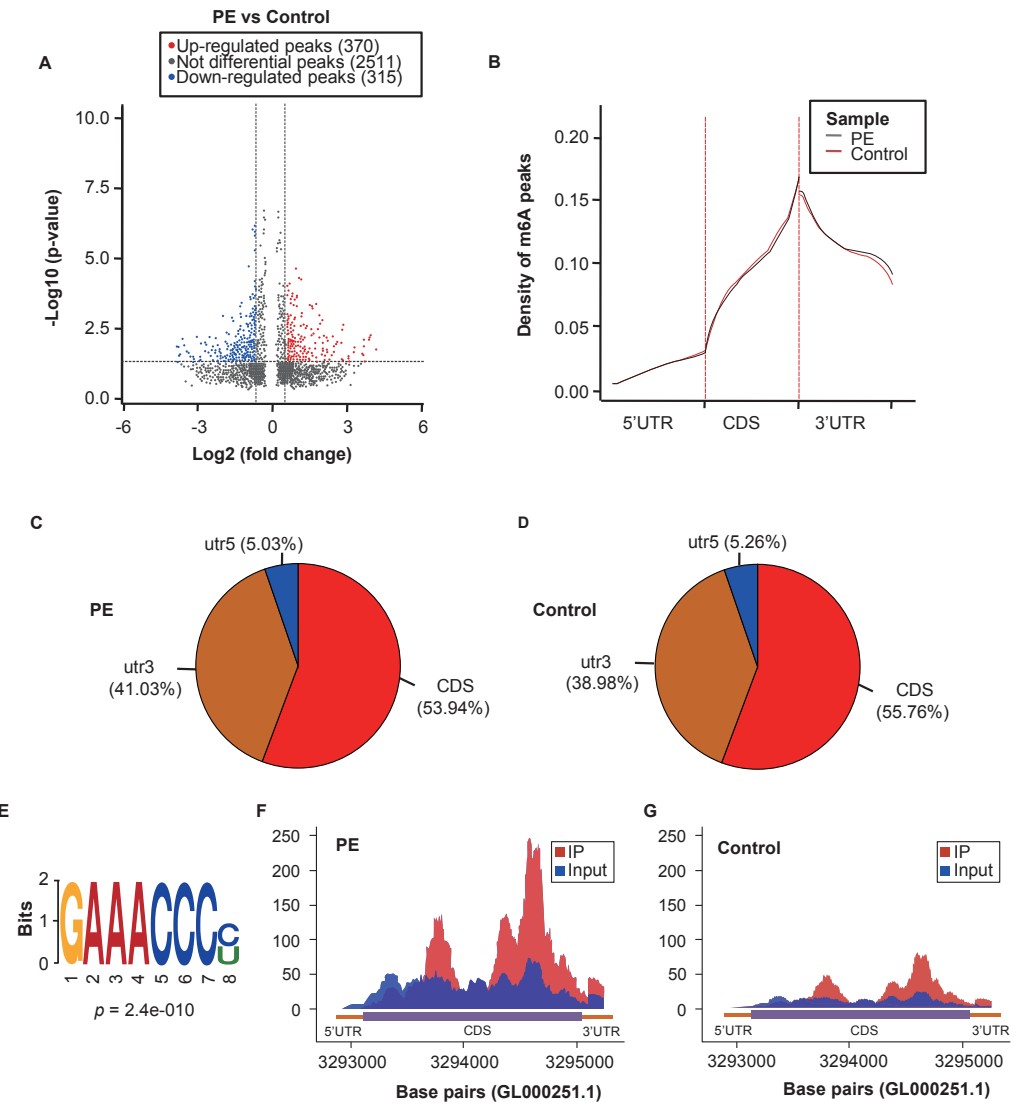

**Figure 3** **Overview of the m6A methylation landscape in the preeclampsia and control samples.** (A) Volcano plots displaying the distinct m6A peaks and their statistical significance (fold changes $\geq 1.5$ and $p < 0.05$). (B) Metagene plots showing the region of average m6A peaks throughout the transcripts in the preeclampsia and control samples. (C) Pie charts displaying the distribution of m6A peaks in the preeclampsia. (D) Pie charts displaying the distribution of m6A peaks in the control group. (E) Sequence motifs of the m6A-containing peak regions. (F) Data visualisation analysis of *HSPA1A* mRNA m6A modifications in the preeclampsia group. (G) Data visualisation analysis of *HSPA1A* mRNA m6A modifications in the control group.

5′GAAGGUGCUGGACAAGUGUCAAGA 3′and 5′GAAGGUGCUGG(m6A)CAAGU-GUCAAGA3′, respectively. Additionally, *HSPA1A* mRNA and HSP70 protein expression was significantly increased in PE (Figs. 6C & 6D; *HSPA1A* mRNA $p = 0.048$).

**Table 3   The top 20 differently expressed m6A peaks in PE.**

| Gene name | Fold change | Regulation | Chromosome | Peak start | Peak end | Peak region | *P*-value |
|-----------|-------------|------------|------------|------------|----------|-------------|-----------|
| HSPA1A | 47.50 | Up | GL000251.1 | 3294734 | 3294974 | cds | 5.13E−06 |
| DMWD | 18.38 | Up | chr19 | 46293970 | 46294029 | cds, utr3 | 1.82E−02 |
| HYOU1 | 15.67 | Up | JH159138.1 | 67181 | 68226 | cds, utr3 | 5.62E−03 |
| BRCA1 | 15.14 | Up | chr17 | 41245740 | 41245861 | cds | 6.61E−03 |
| NDUFB2 | 13.36 | Up | chr7 | 140396821 | 140400705 | utr5 | 4.79E−02 |
| SLC39A1 | 13.09 | Up | chr1 | 153939949 | 153940038 | utr5 | 1.38E−02 |
| HYOU1 | 13.00 | Up | JH159138.1 | 67162 | 68237 | cds, utr3 | 7.94E−03 |
| SCAF11 | 13.00 | Up | chr12 | 46322790 | 46322910 | utr5 | 2.75E−02 |
| SLC25A29 | 12.47 | Up | chr14 | 100760163 | 100760343 | utr5 | 2.63E−02 |
| MID1IP1 | 9.99 | Up | chrX | 38663134 | 38663344 | utr5 | 1.55E−02 |
| SLC6A2 | 0.07 | Down | chr16 | 55689515 | 55689784 | utr5 | 1.41E−02 |
| NPIPB6 | 0.08 | Down | chr16 | 28353905 | 28354053 | utr3, cds | 7.59E−03 |
| LEKR1 | 0.09 | Down | chr3 | 156544207 | 156544506 | utr5 | 1.38E−02 |
| MBLAC2 | 0.09 | Down | chr5 | 89769658 | 89769897 | cds | 2.19E−02 |
| TSSK6 | 0.10 | Down | chr19 | 19626179 | 19626300 | cds, utr5 | 3.16E−02 |
| LRRC3 | 0.10 | Down | chr21 | 45877072 | 45877311 | cds, utr3 | 3.47E−02 |
| RSPO1 | 0.12 | Down | chr1 | 38077963 | 38078322 | utr3 | 4.90E−02 |
| AC016586.1 | 0.13 | Down | chr19 | 4041364 | 4041783 | utr3 | 6.17E−03 |
| TEX40 | 0.14 | Down | chr11 | 64071267 | 64072238 | cds, utr3 | 3.16E−02 |
| RPP25 | 0.15 | Down | chr15 | 75246876 | 75246966 | utr3 | 4.90E−02 |

## DISCUSSION

It has long been accepted that placental structural and functional abnormalities can cause a number of pregnancy-associated diseases such as PE, gestational diabetes mellitus, intrauterine growth restriction, and gestational trophoblastic disease (*Burton & Jauniaux, 2018*; *Burton et al., 2019*; *Chaiworapongsa et al., 2014*; *Mol et al., 2016*; *Salomon et al., 2016*; *Veras et al., 2017*). Portions of the placenta entering systemic circulation cause maternal PE syndrome, which includes oxidative stress of the syncytiotrophoblast and dysregulated uteroplacental perfusion (*Burton et al., 2019*; *Redman, Sargent & Staff, 2014*). PE is also associated with changes in placental DNA methylation (*Herzog et al., 2017*) and gene expression (*Junus et al., 2012*; *Liang et al., 2016*). This study reveals a region-specific m6A methylation map, first, we found that METTL3 and METTL14 were up-regulated in PE as determined by qPCR. Consistent with this result, the total RNA m6A levels were increased in PE. Then, we identified numerous m6A changes in PE placentas compared to healthy controls after excluding the influence of maternal age, body mass index, perinatal complications, and gestational age at diagnosis and delivery. A portion of the genes identified in this study were previously associated with PE; however, we also identified a number of novel genes. Studies of the human placenta not only enrich our understanding of the mechanisms underlying PE but also provide a theoretical basis for the prevention and pre-emptive treatment of related diseases. Recent studies suggest that the placenta may play a specific role in mediating the generation of intergenerational inheritance (*Padmanabhan*

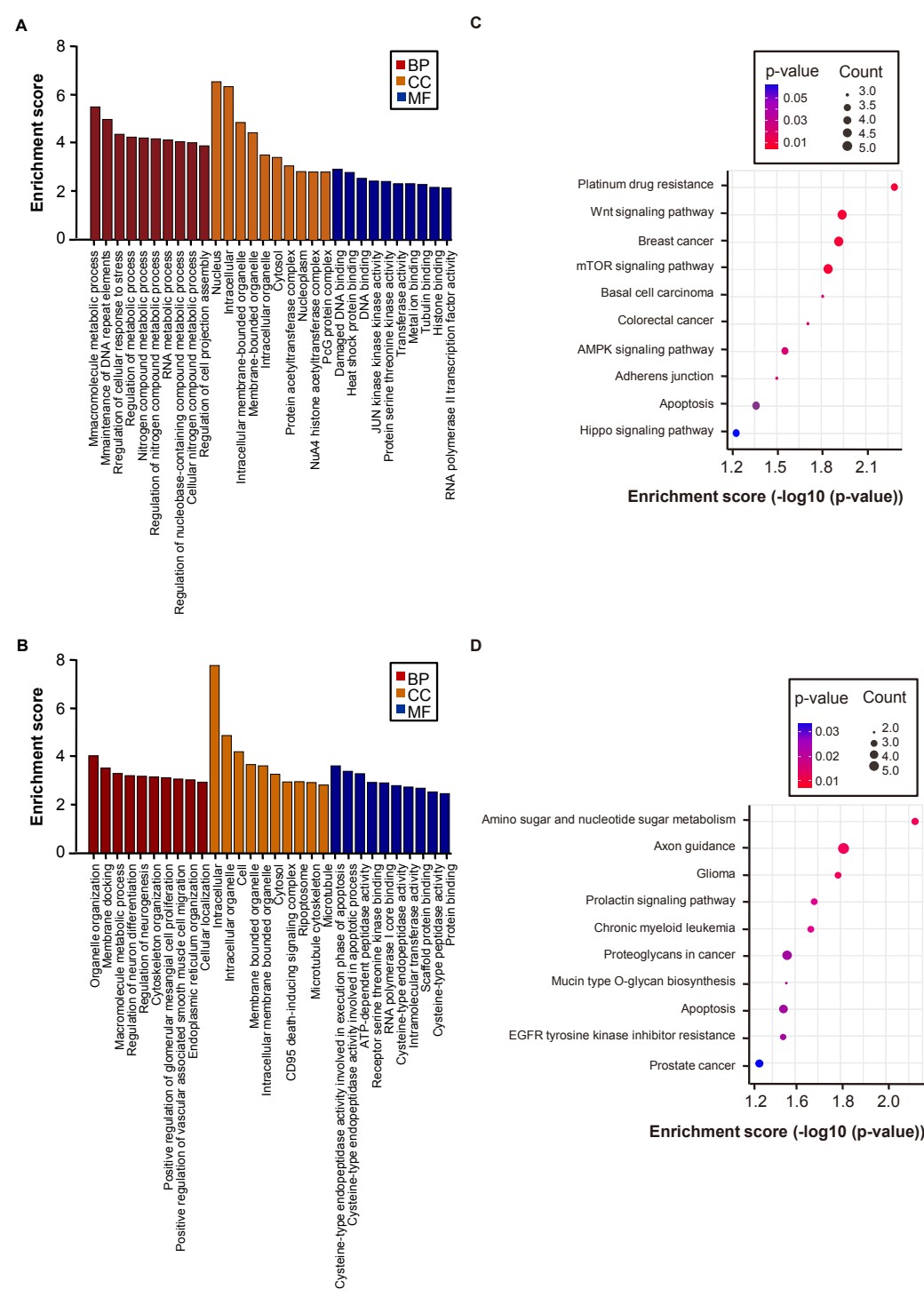

**Figure 4   Gene ontology and KEGG pathway analyses of the altered m6A transcripts.** (A) Major enriched and significant GO-assessed up-regulated m6A peak transcripts. (B) Major enriched and significant GO-assessed down-regulated m6A peaks transcripts. (C) The top ten significantly enriched pathways for the up-regulated m6A peaks transcripts. (D) The top ten significantly enriched pathways for the down-regulated m6A peaks transcripts. GO, gene ontology; KEGG, Kyoto Encyclopedia of Genes and Genomes; BP, biological process; CC, cellular component; MF, molecular function.

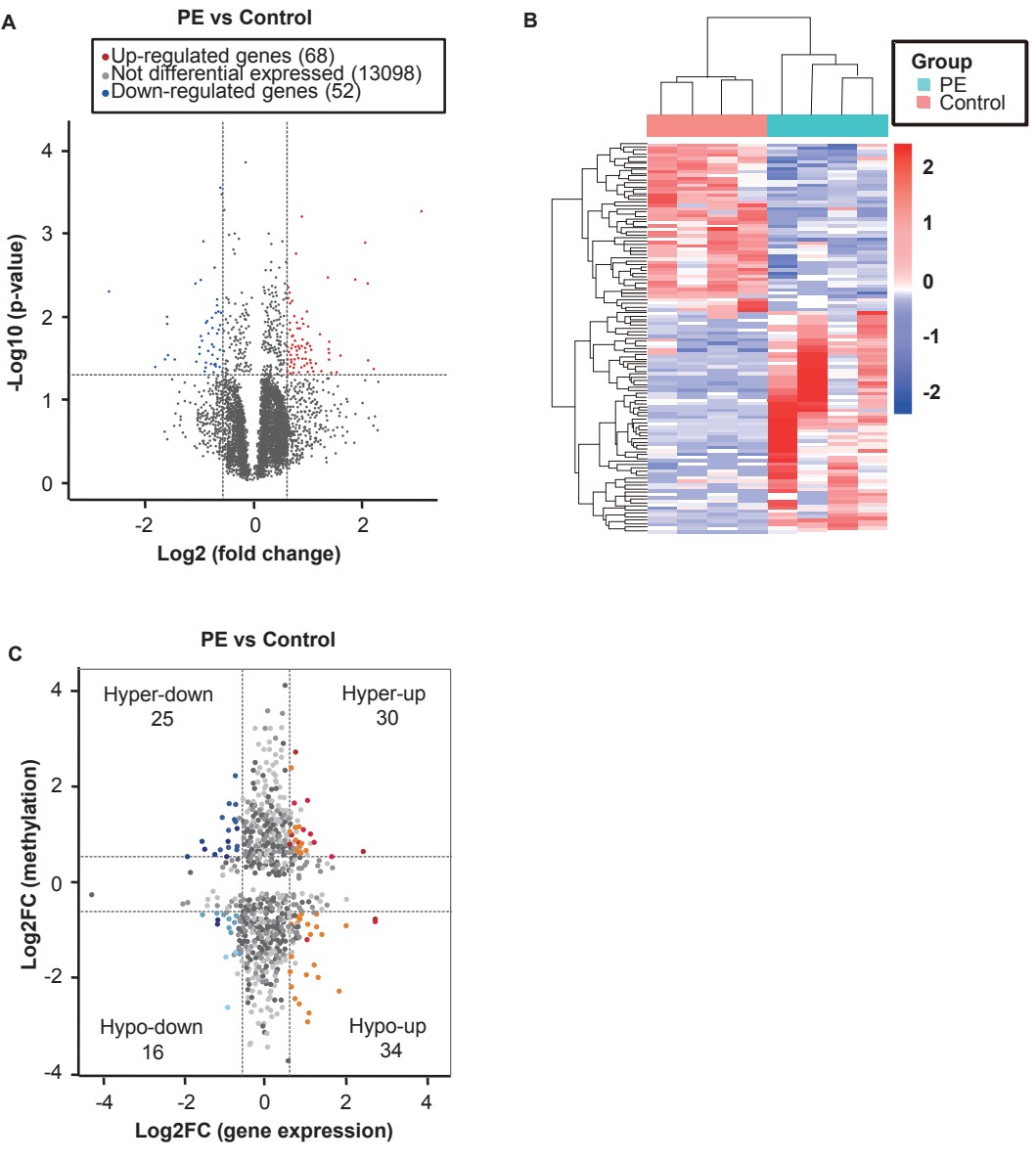

**Figure 5** **Conjoint analysis of m6A-RIP-seq and RNA-sequencing data for preeclampsia and control samples.** (A) Volcano plots displaying the mRNAs that were differentially expressed between the preeclampsia and control groups and their statistical significance (fold changes ≥1.5 and $p < 0.05$). (B) Hierarchical clustering analysis of the differentially expressed mRNAs. (C) Four quadrant graph showing the distribution of transcripts with a significant change in both m6A level and expression in preeclampsia.

*et al., 2013*). Therefore, an in-depth study of the placenta is crucial to determine the mechanisms underlying developmental programming; any related interventions can not only improve maternal and foetal health during pregnancy but also improve the health of multiple generations.

m6A methylation plays a critical role in the regulation of coordinate transcriptional and post-transcriptional gene expression (*Edupuganti et al., 2017*; *Fu et al., 2014*; *Liu et*

**Table 4 The top 20 differently expressed genes in PE.**

| Gene name | Fold change | Regulation | Locus | Strand | p-value |
|---|---|---|---|---|---|
| AOC1 | 8.51 | Up | chr7:150521715-150558592 | + | 5.48E−04 |
| HLA-DRB5 | 4.45 | Up | chr6:32485120-32498064 | − | 4.50E−02 |
| DIO2 | 4.31 | Up | chr14:80663870-80854100 | − | 4.09E−03 |
| NOTUM | 4.27 | Up | chr17:79910383-79919716 | − | 3.34E−02 |
| SPIN1 | 4.04 | Up | chr9:91003334-91093609 | + | 1.26E−03 |
| LAIR2 | 3.55 | Up | chr19:55009100-55021897 | + | 3.46E−03 |
| PLAC8 | 2.96 | Up | chr4:84011201-84058228 | − | 2.89E−02 |
| DCD | 2.61 | Up | chr12:55038375-55042277 | − | 4.97E−02 |
| UPK1B | 2.57 | Up | chr3:118892364-118924000 | + | 2.42E−02 |
| PTPRCAP | 2.57 | Up | chr11:67202981-67205538 | − | 3.48E−03 |
| H2AFJ | 0.67 | Down | chr12:14927317-14930936 | + | 2.06E−02 |
| AHSA2 | 0.67 | Down | chr2:61404553-61418338 | + | 2.87E−02 |
| B3GNT2 | 0.66 | Down | chr2:62423248-62451866 | + | 2.51E−02 |
| PAK1 | 0.66 | Down | chr11:77032752-77185680 | − | 4.96E−02 |
| TMEM14B | 0.66 | Down | chr6:10747992-10852986 | + | 8.71E−03 |
| IMPA2 | 0.66 | Down | chr18:11981024-12030882 | + | 2.68E−04 |
| PKNOX2 | 0.65 | Down | chr11:125034583-125303285 | + | 3.23E−04 |
| SLC9A6 | 0.65 | Down | chrX:135056000-135129428 | + | 1.21E−02 |
| HIST4H4 | 0.63 | Down | chr12:14920933-14924065 | − | 1.74E−02 |
| ANKRD33 | 0.63 | Down | chr12:52281744-52285448 | + | 1.61E−02 |

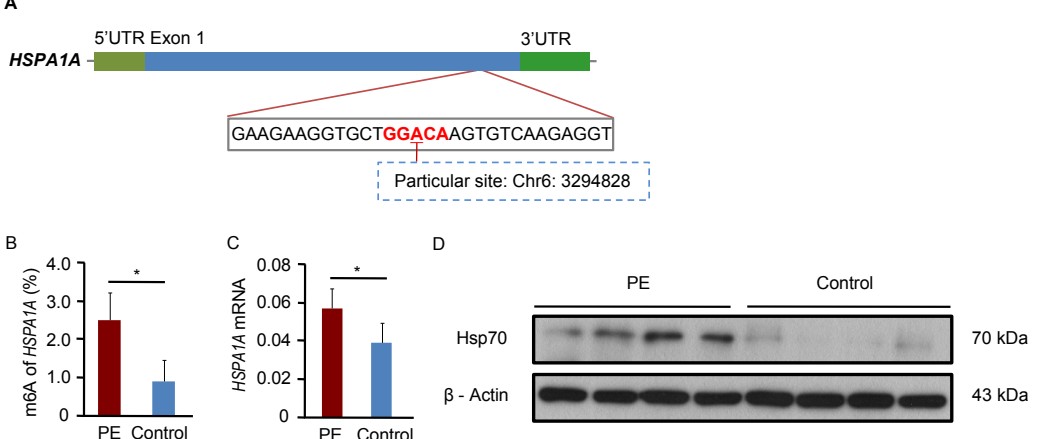

**Figure 6 The expression of HSPA1A is regulated by m6A modification.** (A) Schematic presentation of the location of verified m6A site in *HSPA1A* (B) The m6A levels of *HSPA1A* at the particular site in the CDS region were significantly upregulated in PE. (C) The mRNA levels of *HSPA1A* were significantly increased in PE. (D) HSP70 protein expression was assessed by Western blot. All $p$-values were calculated using Student's $t$-test. $^*p < 0.05$ versus control group ($n = 4$ each). PE, preeclampsia.

*al., 2017a*; *Piao, Sun & Zhang, 2017*; *Wang et al., 2014*; *Yue, Liu & He, 2015*; *Zheng et al., 2017*), including mRNA splicing, export, localisation, translation, and stability. m6A within a transcript can facilitate the binding of regulatory proteins and affect the potential of a given transcript for translation. Recent studies have demonstrated a vital functional role for m6A modification in promoting mRNA translation (*Shi et al., 2017*; *Wang et al., 2015*). In this study, hundreds of abnormal m6A methylations were also found in the transcripts, including within zinc finger transcription factors, which uncovers emerging links between m6A mRNA methylation and genome transcription. Thus, RNA decoration by m6A in these transcripts that promotes their interaction with nuclear transcription factors, possibly conferencing recognition of these transcripts (*Dominissini et al., 2012*).

HSPA1A (heat shock protein 70, Hsp70) is a member of the chaperone machinery that plays a central role in maintaining cellular homeostasis; this protein can elicit innate and adaptive pro-inflammatory immune responses and reflect oxidative stress in PE (*Molvarec et al., 2011*; *Witkin, Kanninen & Sisti, 2017*), which is relevant to the progression of PE (*Chaiworapongsa et al., 2014*). Interestingly, circulating Hsp70 can be directly involved in endothelial activation or injury in PE (*Molvarec et al., 2011*; *Witkin, Kanninen & Sisti, 2017*). Indeed, in this study, we detected a significant increase in HSPA1A mRNA and protein expression in PE. Furthermore, in a previous study, compared with normal pregnancy, Hsp70 was significantly up-regulated in PE placental tissues (*Molvarec et al., 2011*; *Sheikhi et al., 2015*; *Witkin, Kanninen & Sisti, 2017*). However, the exact mechanism of Hsp70 up-regulation in PE remains unclear; it may be related to higher m6A methylation levels of HSPA1A in PE placentas than those in the control samples. IGF2BPs are a distinct family of m6A readers that promote the stability and storage of their target mRNAs and therefore affect gene expression output (*Huang et al., 2018*). Moreover, 92% of the IGF2BP binding sites are located in protein-coding transcripts and highly enriched in CDS near stop codons and in 3'UTRs (*Huang et al., 2018*). Therefore, we concluded that the increase of m6a methylation in the CDS region of HSPA1A may promote mRNA stability. In the present study, we verified that the m6A levels in HSPA1A at the particular site in the CDS region were significantly increased in PE using MazF-qPCR. *Zhou et al. (2015)* reported that the m6A modifications of HSPA1A were increased at 5'UTR in mouse embryonic fibroblast (MEF) cell line after heat shock stress. Compared with this study, the two studies have different species, different tissues, different cells and different treatment process, which may be the reasons for the different results of two studies. Further studies are necessary to elucidate if Hsp70 expression is correlated with m6A methylation of HSPA1A in human placenta.

KEGG pathway analysis revealed that these target genes were significantly enriched in 20 different pathways, including the ''Wnt signalling pathway'', ''mTOR signalling pathway'', ''AMPK signalling pathway''. Some of the most essential and crucial pathological changes in the PE process include shallow trophoblast invasion and the damaged spiral artery remodelling, which may be regulated by Wnt/$\beta$-catenin signalling pathway (*Zhang et al., 2017b*). As a canonical Wnt-signalling pathway, the Wnt/$\beta$-catenin signalling pathway can regulate several biological processes such as proliferation, migration, invasion, and apoptosis (*Nusse & Clevers, 2017*; *Zhang et al., 2017b*). Abnormal functioning of the

Wnt/$\beta$-catenin signalling pathway may play an important role in the pathogenesis of various human diseases, including human cancer and PE (*Nusse & Clevers, 2017*; *Zhang et al., 2017b*); however, direct evidence of the role of the Wnt/ $\beta$-catenin pathway in the development of PE is lacking. In this study, genes exhibiting abnormal m6A methylation were significantly enriched in the Wnt signalling pathway; this may shed light on a new layer of gene regulation at the RNA level, ultimately giving rise to the field of m6A epitranscriptomics. Interestingly, inhibition of Wnt/ $\beta$-catenin signalling is a promising treatment approach for a number of cancers (*Clara et al., 2019*; *Krishnamurthy & Kurzrock, 2018*; *Nusse & Clevers, 2017*). Future studies are recommended to verify if Wnt/ $\beta$-catenin signalling may provide a prospective therapeutic target for the prevention and treatment of PE.

In this study, 120 significantly differentially expressed genes were identified in PE, some genes have been reported in previous studies, including *DIO2*, *BIN2*, *IGSF8*, *BMP5*, and *WNT2* (*Sitras et al., 2009*; *Zhang et al., 2016*; *Zhang et al., 2017b*). These genes, when differentially expressed in different populations, may play an important role in the development of PE. Using bioinformatics analysis, differentially expressed genes were enriched in biological processes involved in defence response, sphingolipid metabolic processes, and tissue development, among others. Maternal-foetal immune incompatibility and disruption of sphingolipid metabolism have emerged as factors involved in the pathogenic mechanisms underlying PE (*Chaiworapongsa et al., 2014*; *Charkiewicz et al., 2017*; *Zhang et al., 2017b*). This provides further evidence that PE is a heterogeneous and multifactorial disease and that a variety of pathogenic mechanisms are implicated in its occurrence and development.

Conjoint analysis of m6A-RIP-seq and RNA-seq data identified m6A-modified mRNA transcripts that were hyper-methylated or hypo-methylated and significantly differentially expressed. *TCF7L2*, a transcription factor of the Wnt and Hippo signalling pathway, is hyper-methylated and up-regulated in PE, suggesting a possible positive relationship between the extent of m6A methylation and the transcript level. Earlier findings revealed that a *TCF7L2* variant increased the risk of incident hypertension or diabetes mellitus (*Bonnet et al., 2013*; *Chang et al., 2017*); however, the relationship between *TCF7L2* and PE is not clear. To provide further insight, it will be necessary, in a future study, to elucidate the biological function of *TCF7L2* in the context of PE.

Gestational age is an important confounder when studying variation in placental DNA methylation. Previous studies have illustrated gestational age as one of the important factors that influence methylation (*Novakovic & Saffery, 2012*; *Wilson et al., 2018*). Placentas obtained from preterm pregnant women with early PE were used in some studies, to match gestational age, some placentas of women undergoing spontaneous preterm delivery were used as control groups, which may possess additional differences in methylation that are not detected at full-term ($\geq$37 week gestation) (*Novakovic & Saffery, 2012*; *Wilson et al., 2018*; *Yeung et al., 2016*). In this study, to exclude the influence of gestational age, we used full-term placentas obtained from women with PE; however, we did not perform a study to determine that m6a methylation changes with gestational age. This is a limitation of the present study.

## CONCLUSIONS

Here, we describe, for the first time, the m6A RNA methylation landscape; reveal the potential functions of this methylation in the regulation of RNA metabolism in PE. Additionally, conjoint analysis of m6A-RIP-seq and RNA-seq data resulted in the identification of differentially expressed hyper-methylated or hypo-methylated mRNA m6A peaks. Finally, we uncovered that *HSPA1A* might be involved in the pathophysiology of PE as its mRNA and protein expression is regulated by m6A modification. Further studies will be necessary to validate more m6A-enriched genes and examine protein levels of the genes which contribute to elucidate the detailed molecular mechanism underlying the regulation and biological functions of m6A during the PE process. Additionally, characterisation of the role of m6A within the placenta will offer a new viewpoint to elucidate the mechanism of PE.

### Funding

This research was funded by the Natural Science Foundation of Shandong Province (No. ZR2014HP047), the Science and Technology Development Project of Jinan (No. 201907013), and the Science and Technology Project of Jinan Health Committee (No. 2019-1-33). The funders had no role in study design, data collection and analysis, decision to publish, or preparation of the manuscript.

### Grant Disclosures

The following grant information was disclosed by the authors:
Natural Science Foundation of Shandong Province: ZR2014HP047.
Science and Technology Development Project of Jinan: 201907013.
Science and Technology Project of Jinan Health Committee: 2019-1-33.

### Competing Interests

The authors declare there are no competing interests.

### Author Contributions

- Jin Wang conceived and designed the experiments, performed the experiments, prepared figures and/or tables, authored or reviewed drafts of the paper, and approved the final draft.
- Fengchun Gao conceived and designed the experiments, prepared figures and/or tables, authored or reviewed drafts of the paper, and approved the final draft.
- Xiaohan Zhao analyzed the data, prepared figures and/or tables, and approved the final draft.
- Yan Cai analyzed the data, prepared figures and/or tables, authored or reviewed drafts of the paper, and approved the final draft.
- Hua Jin conceived and designed the experiments, authored or reviewed drafts of the paper, and approved the final draft.

## Human Ethics

The following information was supplied relating to ethical approvals (i.e., approving body and any reference numbers):

The study protocols were approved by the Ethics Review Committee of Ji'nan Maternal and Child Health Care Hospital and conducted in accordance with the Declaration of Helsinki (No. JNFY-2019003).

## Data Availability

The sequencing data are available in the Supplemental Files and also available at the Gene Expression Omnibus (GEO): GSE143966 and GSE143953.

## Supplemental Information

Supplemental information for this article can be found online at http://dx.doi.org/10.7717/peerj.9880#supplemental-information.

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
