# Peer review of "Integrated analysis of the transcriptome-wide m6A methylome in preeclampsia and healthy control placentas"

_PeerJ, doi:10.7717/peerj.9880_

## Round 0.1 · original submission · Major Revisions

The major concerns are the small sample size and the lack of verifications. I recommend the authors perform additional experiments to address issues raised by reviewers.

·

Basic reporting

The hypothesis that “to speculate that deregulation of m6A modification may also be
associated with PE” is acceptable. But Taniguchi et al. have been already published the role of m6A in PE (Taniguchi et al. PMID: 31914637. FASEB J. 2020 Jan;34(1):494-512. doi: 10.1096/fj.201900619RR.). The authors should refer this article.

Experimental design

The number of subjects is small. The authors should validate their results as adding validation subjects and examine protein levels of the genes which differentially expressed m6A peak in PE.

Validity of the findings

The functional meanings of m6A modifications is differed by its site in transcript and by which m6A reader that recognize m6A modifications bind on those sites. Hence, the authors should provide annotations of m6A peak regions in Table 3 such as 5’UTR, the beginning, the middle, and the last part of CDS, and 3’UTR.

The authors should provide not only the top 20 differentially expressed m6A peaks in PE but also whole differentially expressed m6A peaks data as supplementary data.

In conclusion, “the m6A RNA methylation landscape; reveal the potential functions of this methylation in the regulation of RNA metabolism” is based on results. However, “various biological processes in PE that may provide promising therapeutic targets for the prevention and treatment of PE.” is a rough conclusion, because it is unclear whether the modifications by methylation enhance protein levels or not. As mentioned above, authors should check changes in protein levels of differentially methylated transcripts.

Additional comments

In the present manuscript entitled “Integrated analysis of the transcriptome-wide m6A
methylome in preeclampsia and healthy control placentas”, Wang et al. explored placental molecular pathology of preeclampsia (PE) by introducing MeRIP-Seq. It revealed that PE accompanied post-transcriptional changes in several genes with modifying adenosine to N6-mehyladenosine or vice versa in their transcripts. These results suggest potential function of m6A modification in incidence of PE. However, there are some critical points as follows.

Major comments
1. The sample size in this study is too small to conclude. In addition, the eventual m6A modification’s effects on gene expression is unknown. It is published that HSPA1A transcript increased m6A modifications at 5’ UTR during heat shock stress and which augment protein translation (Zhou et al. PMID: 26458103. Nature 2015 Oct 22;526(7574):591-4). Therefore, the authors should validate their results in protein levels as adding samples. Further, the authors should show the position of peak summit and in which transcript region (5’UTR, the beginning, the middle, and the last part of CDS, and 3’UTR) is located in peak region in table 3.
2. The effects of methylation on transcripts fate depends on which m6A reader protein binds to m6A sites. The authors should also show the expression levels of m6A readers, writers, and erasers in each group from RNA-Seq results.
3. Is there any contents in showing the m6A peak changes for each chromosome in Fig 3D? Are there significant differences compared to expression changes?
4. Which genes were included in hyper-methylated and down-regulated, and hypo-methylated and up-regulated?
5. The authors should provide mean and SD of m6A peaks height in each region of each group in addition to fold changes as for significantly different m6A peaks between PE and control.
6. PE subjects delivered larger babies than normal subjects. It differs from the sample characters in Taniguchi et al.’s report. When did PE subjects in this study develop PE?


Minor comments
1. Line 95. Was the mRNA mixed from the three different sites of placenta from an identical subject for MeRIP-Seq?
2. Which version of Ensembl reference genome were referred for mapping?
3. The authors fragmented RNA into 100-nt fragments for M6A-RIP. But sequencing was performed on pair-end 300 cycles protocol. It means one fragment was read twice from both directions. The authors should use single-end sequence data for mapping and peak detection.
4. Line 207. Which value was adapted for cut off in filtering significant enrichment in GO analysis?
5. Fig 3B and F. Which software did authors use for plot?
6. “P” is missing in the title of Figure 3 and 4 for “preeclampsia”.

Reviewer 2 ·

Basic reporting

no comment

Experimental design

Strengths:
Research questions is clearly defined with an identified knowledge gap.

Reasonable biological replicates and comparison groups included in the study.

Weaknesses:
Lack of orthogonal validation of sequencing results is a major weakness.

Validity of the findings

Weaknesses:
Key information/data, such as sequencing depth and sequencing data availability, are missing.

There are not validations for the sequencing results, which makes it difficult to evaluate the robustness of the analysis data and related biological interpretation.

---

## Round 0.2 · Major Revisions

The major issues in the initial submission still remain in the revision. The authors should perform additional analyses to validate the conclusions.

·

Basic reporting

The profile of RNA m6A modification in PE placenta could be a breakthrough to reveal the mechanism of PE. However, there is a lack of verifications.

Experimental design

The sample size is small.

Validity of the findings

There are no additional experiments to verify their results or hypothesis.

Additional comments

The knowledge about RNA m6A modification is now widely accumulated in various samples. Therefore, subsequent verification is a necessity to accept this manuscript. I regret that the authors did not perform additional experiments.

---

## Round 0.3 · Minor Revisions

The reviewer has been helpful and constructive for increasing the quality of the manuscript. Please, read the critique carefully and address all items.

·

Basic reporting

In abstract, authors described "However, the profiling of m6A modification and its potential
role in preeclampsia (PE) has not yet been studied." But there is a report regarding to profiling of m6A in PE placentas. It should be described like "The number of reports regarding to the profiling of m6A modification and its potential role in the placenta of preeclampsia (PE) is a little.

The authors should show the "particular site" of HSPA1A by genomic locus, and also should indicate it by an arrow in Fig 3G.

The annotations for Fig 3 is not matched in the manuscript.

There is no plot in Fig 4C.

Experimental design

As pointed out in the above section, this is not a first report that profiling of PE placentas were performed. The authors should make clear their original research aims.

line 201 to 207.
First, the authors should show the quality of their extracted RNA. Next, why have the authors show the results of only qRT-PCR even though they have done RNA-Seq?
The authors should show the expression levels by RNA-Seq results, too, regarding to the genes in Figure 1. When describing about gene expression, the gene names should be in italic.

Regarding to conjoint analysis, which category did the HSPA1A belong? There is no HSPA1A in table S3 although the authors show the upregulation o HSPA1A mRNA levels in Figure 6B. And what do the authors think about the final effects of each category to translation?

Validity of the findings

The authors should show the results of m6A peaks in gene level, too. How many genes have both up- and down- regulated m6A peaks? How many modified peaks are there in one gene? When performing conjoint analysis, how did the authors divide the transcripts that both up- and down- regulated m6A peaks exist?

Why are the authors interested in the distribution of altered m6A peaks by chromosomes?What does it mean concerning m6A modifications? There is no consideration regarding to this result. The result of Fig 3E is not needed in this manuscript.

Regarding to Fig 3G and 3H, the authors should show the map of HSPA1A transcript so that the readers can know where are UTRs and where is the stop codon. In addition, the authors should show the peaks of HSPA1A in every 8 sample (4 PE and 4 Control). Furthermore, it seems there are two different blue colors in Fig 3G and H.

Regarding to Fig 5C, what relationship exist between m6A modification and the novel transcripts possessed coding capabilities? There is no additional experiments regarding these transcripts. The result of Fig 5C is not needed in this manuscript.

How does it work to final output when the m6A modification of "particular" HSPA1A changed from 1% to 2.5% of total in Fig 6A? There is a dissociation between transcript levels and m6A % at "particular" site and protein levels.

Additional comments

line 275 to 276, the authors have not investigated "the role of placental m6A in PE" in this manuscript. The effects of alteration of m6A appear to be nothing more than speculation except HSPA1A. Therefore, it is overestimation. As well as the same reason, the results of GO or KEGG is just annotation. We do not know whether these pathways are activated or repressed eventually in PE placentas. Therefore the volume of discussion regarding to the results of GO or KEGG is too much.

---

## Round 0.4 · accepted · Accept

I suggest minor changes (see attached file).